# Extracellular matrix sensing by FERONIA and Leucine-Rich Repeat Extensins controls vacuolar expansion during cellular elongation in *Arabidopsis thaliana*

Kai Dünser[1] iD, Shibu Gupta[2] iD, Aline Herger[2] iD, Mugurel I Feraru[1], Christoph Ringli[2] iD & Jürgen Kleine-Vehn[1,*] iD

## Abstract

**Cellular elongation requires the defined coordination of intra- and extracellular processes, but the underlying mechanisms are largely unknown. The vacuole is the biggest plant organelle, and its dimensions play a role in defining plant cell expansion rates. Here, we show that the increase in vacuolar occupancy enables cellular elongation with relatively little enlargement of the cytosol in *Arabidopsis thaliana*. We demonstrate that cell wall properties are sensed and impact on the intracellular expansion of the vacuole. Using vacuolar morphology as a quantitative read-out for intracellular growth processes, we reveal that the underlying cell wall sensing mechanism requires interaction of extracellular leucine-rich repeat extensins (LRXs) with the receptor-like kinase FERONIA (FER). Our data suggest that LRXs link plasma membrane-localised FER with the cell wall, allowing this module to jointly sense and convey extracellular signals to the cell. This mechanism coordinates the onset of cell wall acidification and loosening with the increase in vacuolar size.**

**Keywords** cell expansion; cell wall; growth; plant; vacuole
**Subject Categories** Membrane & Intracellular Transport; Plant Biology
The EMBO Journal (2019) 38: e100353

## Introduction

Primary plant cell walls are composed of the polysaccharides cellulose, hemicelluloses and pectin along with structural proteins. The extracellular matrix features considerable tensile strength, withstanding the internal hydraulic turgor pressure of cells. This stiff construction provides stability to the plant body, yet obviously conflicts with cellular enlargements. Remarkably, acidic pH of the apoplast (extracellular space) allows cell walls to extend and accordingly coincides with cellular elongation (Barbez *et al*, 2017). Nuclear signalling of the phytohormone auxin is crucial for the control of extracellular pH and plant cell expansion in a concentration- and tissue-dependent manner (Spartz *et al*, 2014; Fendrych *et al*, 2016; Barbez *et al*, 2017). In hypocotyls, the auxin-induced SMALL AUXIN UP RNA 19 (SAUR19) inhibits PP2C-D phosphatases, thereby stimulating the activity of plasma membrane $H^+$-ATPases, which promote cellular expansion in aerial tissues (Spartz *et al*, 2014). Expansins and other cell wall remodelling enzymes have been proposed as mediators of this acid growth mechanism. Developmentally defined cell wall loosening and concomitant water uptake are important pre-requisites to enable turgor-driven cell expansion (reviewed in Braidwood *et al*, 2013). Accordingly, cellular elongation requires a complex coordination of several internal and external processes; however, relatively little is known how cell wall changes are molecularly sensed and translated into intracellular signals. Receptor kinases are predestined to have a role in linking the intra- and extracellular compartments. THESEUS1 (THE1) attenuates defects in cellulose deficiency, proposing a function of *Catharanthus roseus* receptor-like kinase 1-likes (*Cr*RLK1Ls) in cell wall sensing (Hématy *et al*, 2007). In agreement, the receptor kinase *Cr*RLK1L FERONIA (FER) also controls the extensibility of the cell wall (Höfte, 2015) and functions as a mechano-sensor (Shih *et al*, 2014). *Cr*RLK1Ls are transmembrane proteins, typically with an extracellular domain, consisting of two adjacent malectin-like domains, for signal perception and a cytoplasmic kinase domain for intracellular signal transduction (Escobar-Restrepo *et al*, 2007). Besides its contribution to mechanical sensing of the extracellular space, FER is a versatile growth integrator in plants with important contributions to a wide range of responses (reviewed in Li *et al*, 2016). The malectin domains bind to the peptide ligand RAPID ALKALINISATION FACTOR1 (RALF1), which subsequently triggers the FER kinase-dependent repression of cellular expansion

1   Department of Applied Genetics and Cell Biology, University of Natural Resources and Life Sciences (BOKU), Vienna, Austria
2   Department of Plant and Microbial Biology, University of Zurich, Zurich, Switzerland
    *Corresponding author. Tel: +43 1 47654 94150; E-mail: juergen.kleine-vehn@boku.ac.at

(Haruta *et al*, 2014). However, the importance of RALF peptides for the FER-dependent mechano-sensing is unknown.

Besides cell wall loosening and turgor pressure, the dimension of the biggest plant organelle, the vacuole, is another determinant for cellular enlargement. The size of the vacuole correlates with cell size in plants and is controlled by the phytohormone auxin, thereby impacting cellular elongation rates (Löfke *et al*, 2015; Scheuring *et al*, 2016). Despite its eminent importance, it remains largely unknown how intracellular processes, such as the regulation of vacuolar size, are coordinated with cell wall loosening to ensure the coordination of cellular elongation in *Arabidopsis thaliana*. Here, we show that extracellular leucine-rich repeat extensin (LRX) proteins contribute to cell wall sensing via interaction with FER. Our work proposes that LRX proteins bridge the plasma membrane-localised FER with the cell wall, enabling it to sense wall rigidity. This LRX/FER-dependent mechanism conveys extracellular signals to the underlying cell and thereby suppresses growth-relevant processes, such as the intracellular expansion of the vacuole.

## Results

Here, we use epidermal atrichoblast (non-root hair forming) root cells to record vacuole expansion during cellular elongation. To assess the relative dilation of the vacuole, we combined the fluorescent dye BCECF (2′,7′-Bis-(2-carboxyethyl)-5-(and-6)-carboxyfluorescein), which accumulates in the vacuolar lumen of plant cells, with propidium iodide, which stains the exterior of the cells

(Scheuring *et al*, 2015). Subsequently, we performed defined z-stack imaging and 3D rendering of the cell (Movies EV1–EV4), allowing quantification of how much cellular space is filled by the vacuole (also defined as vacuolar occupancy of the cell; Scheuring *et al*, 2016). In young/early meristematic cells, the vacuole occupied about 30–40% of the cellular space (Fig 1A and B; Movies EV1 and EV2). By contrast, the size of the vacuole and its relative occupancy of the cell dramatically increased during cellular elongation, ultimately taking up 80–90% of the cellular volume (Fig 1A and B; Movies EV3 and EV4). While the epidermal cells enlarged their volume by a factor of approximately 14, the absolute space between the vacuole and the cell boundary increased only about twofold to threefold (Fig 1C and Appendix Fig S1A and B). The 3D imaging of a cytosolic fluorescent protein also confirmed that the cytosol shows relatively minor volume expansion during epidermal elongation (Appendix Fig S1A and B). We illustrate that the relative increase in vacuolar size has a dramatic impact on cytosol homeostasis, consequently requiring relatively little *de novo* production of cytosolic components during cellular enlargement. Moreover, we propose that the vacuolar size is a suitable intracellular marker for cellular expansion dynamics.

Cell wall acidification is central in activating a cascade of events, ultimately leading to cell wall loosening and subsequent cellular elongation (Fendrych *et al*, 2016; Barbez *et al*, 2017). Accordingly, the cell wall acidification/loosening in elongating cells (Barbez *et al*, 2017) is coinciding with growth-relevant intracellular processes, such as the increase in vacuolar size (Fig 1A–C). Here, we used the vacuolar size as a suitable intracellular read-out to study the

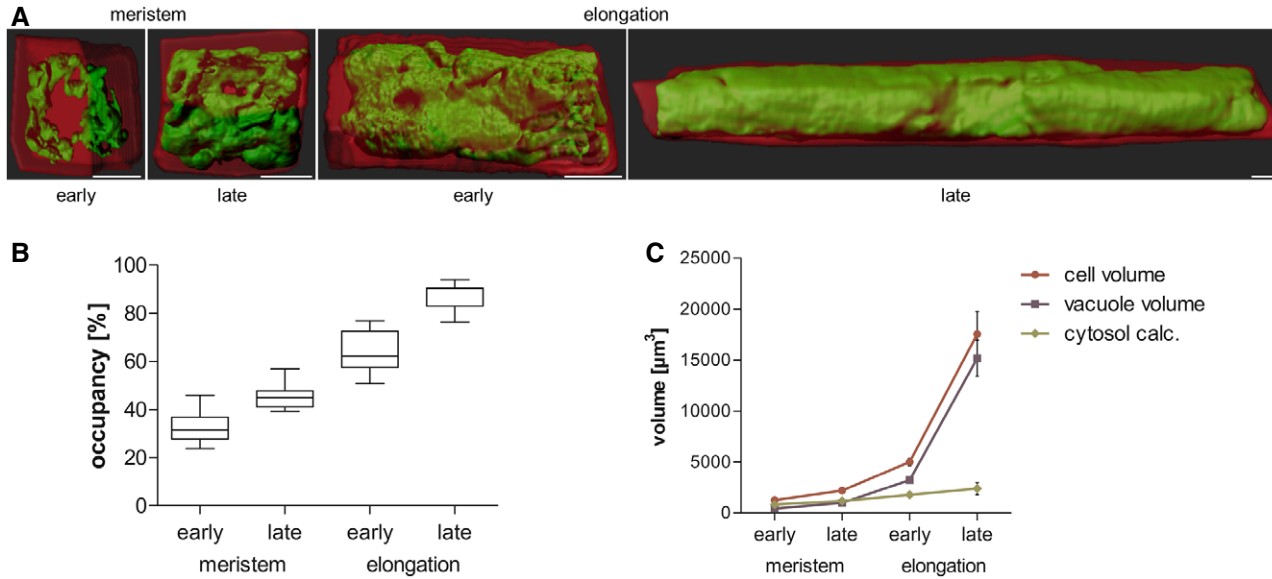

**Figure 1.  Vacuolar occupancy of the cell enables cytosol homeostasis during rapid growth.**

A   3-D reconstructions of propidium iodide (PI)-stained cell walls (red) and BCECF-stained vacuoles (green) of epidermal atrichoblasts in the early and late meristem and in the early and late elongation zone. Scale bars: 5 μm.

B   Boxplots showing vacuolar occupancy of cells in the defined zones (*n* = 7–11). Box limits represent 25[th] percentile and 75[th] percentile; horizontal line represents median. Whiskers display min. to max. values.

C   Graph depicts absolute values of cell volume and vacuolar volume in the depicted root zones (data points display mean, error bars depict s.e.m., *n* = 7–11). "cytosol calc." is approximated as the difference between cell volume and vacuolar volume.

Source data are available online for this figure.

coordination of extracellular and intracellular processes during cellular elongation. We investigated the possibility that apoplast acidification/cell wall loosening is sensed and provides a feedback for vacuolar morphogenesis. We either directly acidified the rhizosphere (by lowering the pH of the media) or genetically (SAUR19 induction) as well as pharmacologically (fusicoccin treatment) induced the activity of the plasma membrane $H^+$-ATPase (Marre, 1979; Spartz *et al*, 2014, 2017). All these conditions have recently been shown to acidify the pH of epidermal cell walls, consequently inducing cellular elongation (Barbez *et al*, 2017). We quantified the vacuolar morphology, depicting the dimension of the biggest

luminal vacuolar structure (Löfke *et al*, 2015; Scheuring *et al*, 2016), in root epidermal cells of the late meristematic zone. This allows us to assess the impact of cell wall acidification before the actual onset of elongation. Using the tonoplast stain MDY-64 or the tonoplast marker line *pUBQ10::VAMP711-YFP* (Geldner *et al*, 2009; Löfke *et al*, 2015; Scheuring *et al*, 2015), we revealed that cell wall acidification correlated with a dramatic alteration in vacuolar morphology (Fig 2A, C and E), leading to an increased vacuolar occupancy of the cell (Fig 2B, D and F). Fusicoccin-induced cell wall acidification had an immediate (within 30 min) influence on the vacuolar lumen (Appendix Fig S2A and B), suggesting that cell wall

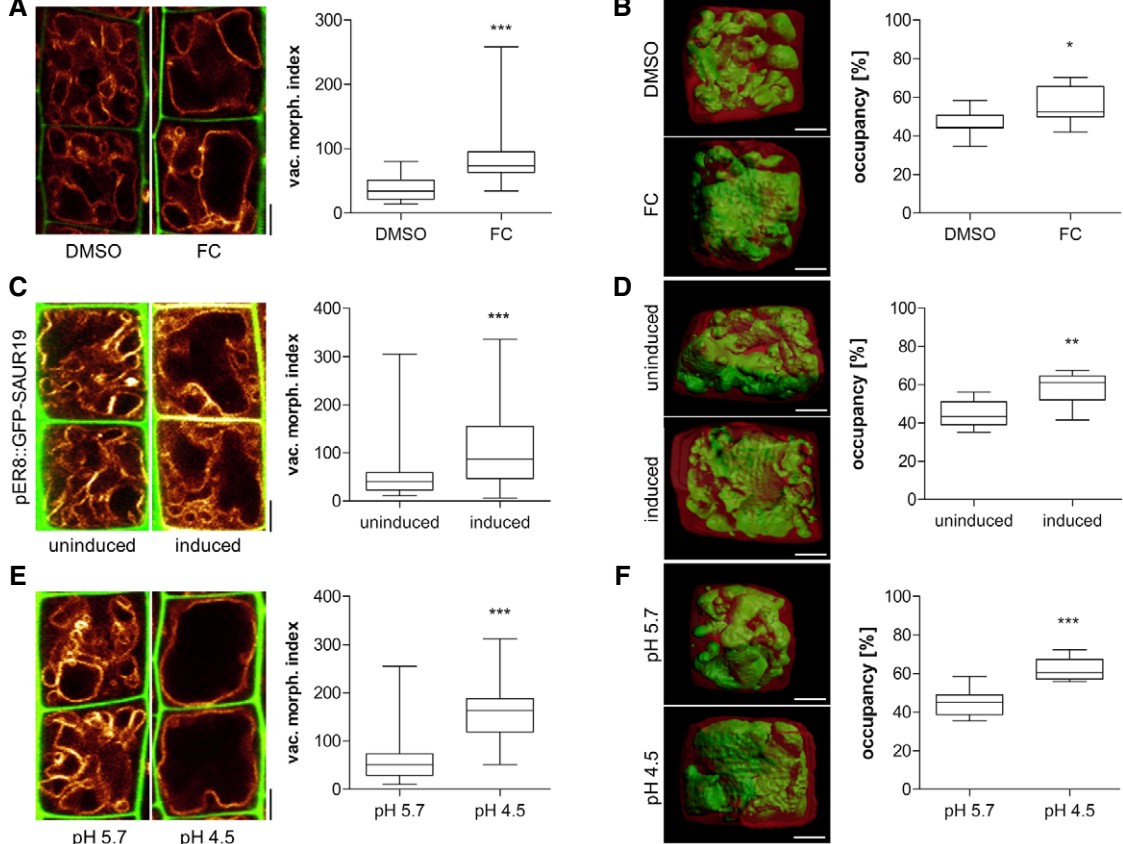

**Figure 2. Vacuolar size correlates with cell wall modifications.**

A   Representative images and quantification of vacuolar morphology of late meristematic cells. PI (green) and *pUBQ10::VAMP711* (yellow) depict cell wall and tonoplast, respectively. Seedlings were treated with DMSO (solvent control) or 5 µM FC (Fusicoccin) for 2.5 h in liquid medium (*n* = 24). Mann–Whitney *U*-test (****P* < 0.001).

B   Left: 3-D reconstructions of PI-stained cell wall (red) and BCECF-stained vacuole (green) of late meristematic cells. Right: Boxplot depicts vacuolar occupancy of the cell treated with the solvent DMSO or 5 µM FC for 2.5 h in liquid medium (*n* = 11). Student's *t*-test (**P* < 0.05).

C   Cell wall and vacuolar membrane were visualised with PI (green) and MDY-64 (yellow). *pER8::GFP-SAUR19* seedlings were treated with DMSO (*n* = 60) or 10 µM β-estradiol (*n* = 56) for 6 h in liquid medium. Mann–Whitney *U*-test (****P* < 0.001).

D   Left: 3-D cell reconstructions of PI-stained cell wall (red) and BCECF-stained vacuole (green) of late meristematic cells in *pER8::GFP-SAUR19* lines. Right: Boxplot depicts vacuolar occupancy of the cell. Seedlings were treated with the solvent control DMSO (*n* = 11) or 10 µM β-estradiol (*n* = 8) for 6 h in liquid medium. Student's *t*-test (***P* < 0.01).

E   Cell wall and vacuolar membrane were visualised with PI (green) and *pUBQ10::VAMP711* (E) (yellow). Col-0 wild-type seedlings were treated for 3 h in liquid medium adjusted to pH 5.7 (*n* = 44) or pH 4.5 (*n* = 40). Mann–Whitney *U*-test (****P* < 0.001).

F   Left: 3-D reconstructions of PI-stained cell wall (red) and BCECF-stained vacuole (green) of late meristematic cells. Right: Boxplot depicts vacuolar occupancy of cell. Seedlings were treated for 3 h in liquid medium adjusted to pH 5.7 or pH 4.5 (*n* = 11). Student's *t*-test (****P* < 0.001).

Data information: Scale bars: 5 µm. Boxplots: Box limits represent 25th percentile and 75th percentile; horizontal line represents median. Whiskers display min. to max. values. Representative experiments are shown.

Source data are available online for this figure.

acidification/loosening directly impacts on vacuolar morphogenesis and size.

The inhibition of pectin methyl esterases (PMEs) leads to reduced cellular elongation, presumably due to stiffening of the cell walls (Wolf et al, 2012). Therefore, we used epigallocatechin gallate (EGCG), which is a natural inhibitor for PMEs (Lewis et al, 2008), to inducibly interfere with the properties of the cell wall. The application of EGCG-induced smaller vacuolar structures and 3D imaging revealed an overall reduced vacuolar occupancy of the cell (Fig 3A and B). Similarly, roots that penetrate a relatively stiff medium also showed smaller vacuoles, when compared to surface-grown roots (Fig 3C and D). These findings suggest that extracellular constraints restrict intracellular enlargements of vacuoles.

We assume that cell wall properties are sensed, which generates a signal to subsequently modulate vacuolar size. To test this hypothesis, we focused on FER receptor-like kinase, which is required for mechanical cell wall sensing (Shih et al, 2014; Feng et al, 2018). Compared to wild-type seedlings, the fer-2 and fer-4 loss-of-function mutants showed enlarged, roundish vacuoles (Fig 4A; Appendix Fig S3A) and increased vacuolar occupancy of the epidermal cells (Fig 4B; Appendix Fig S3B). Notably, epidermal cell length was tendentially slightly enlarged in the root meristem of fer mutant when compared to wild type (Appendix Fig S3C). Importantly, fer

mutant vacuoles were markedly less affected by EGCG treatments or by extracellular constraints of the substrate (Fig 4C and D). In agreement, fer-4 mutants were insensitive to the root growth inhibitory effect of EGCG when compared to wild type (Appendix Fig S3D and E). Accordingly, we conclude that an extracellular, FER-dependent signal impacts intracellular expansion of the vacuole. Notably, an engineered fer mutant, carrying a point mutation in the intracellular kinase domain, was not able to fully complement the vacuolar phenotype of fer4 mutants (Appendix Fig S3F). These data support a role for the FER kinase activity and, hence, FER-dependent signalling in restricting intracellular expansion of the vacuole.

We, subsequently, used the vacuolar morphology as a quantitative read-out to further our knowledge on FER-dependent cell wall sensing mechanisms and turned our attention to extracellular proteins with a possible role in cell wall sensing. Interestingly, leucine-rich repeat extensins (LRXs) are extracellular proteins (Baumberger et al, 2003a) and showed co-expression with FER and THE1 as well as several RALF peptides (Appendix Fig S4A and B). LRX proteins display an N-terminal leucine-rich repeat (LRR) and a C-terminal extensin (EXT) domain (Draeger et al, 2015). While the LRR domain is presumably involved in protein–protein interactions, the EXT domain allows the LRX proteins to bind cell wall components (Ringli, 2010). This domain structure also envisioned the

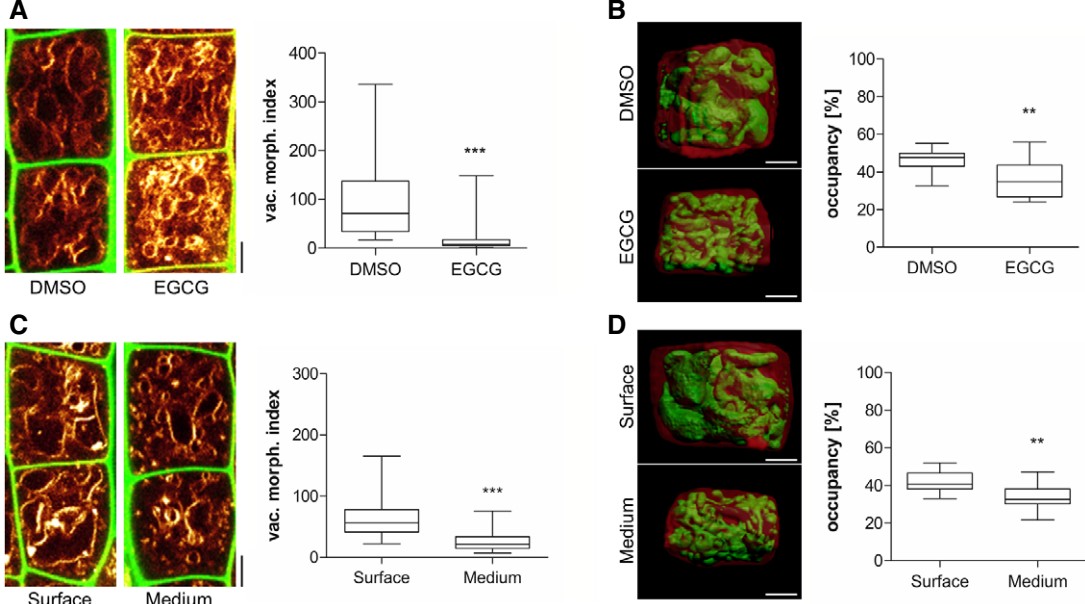

**Figure 3.  Extracellular constraints impact on vacuolar appearance.**

A  Representative images and quantification of vacuolar morphology of late meristematic cells. PI (green) and MDY-64 (yellow) staining depict cell wall and vacuolar membrane, respectively. Seedlings were treated with DMSO solvent control or 50 μM EGCG for 22 h on solid medium (n = 40). Mann–Whitney U-test (***P < 0.001).

B  3-D reconstructions of PI-stained cell wall (red) and BCECF-stained vacuole (green) of late meristematic cells. Boxplot depicts vacuolar occupancy of the cell. Seedlings were treated with DMSO (n = 16) or 50 μM EGCG (n = 15) for 22 h on solid medium. Student's t-test (**P < 0.01).

C  Representative images and quantification of vacuolar morphology of late meristematic cells. PI (green) and MDY-64 (yellow) staining depict cell wall and vacuolar membrane, respectively. Seedling roots were grown on the surface (n = 28) or into the matrix (n = 28) of 2% agar-containing solid medium. Mann–Whitney U-test (***P < 0.001).

D  3-D reconstructions of PI-stained cell wall (red) and BCECF-stained vacuole (green) of late meristematic cells. Boxplot depicts vacuolar occupancy of surface (n = 10) and into the medium (n = 11) grown seedling roots. Student's t-test (**P < 0.01).

Data information: Scale bars: 5 μm. Boxplots: Box limits represent 25th percentile and 75th percentile; horizontal line represents median. Whiskers display min. to max. values. Representative experiments are shown.
Source data are available online for this figure.

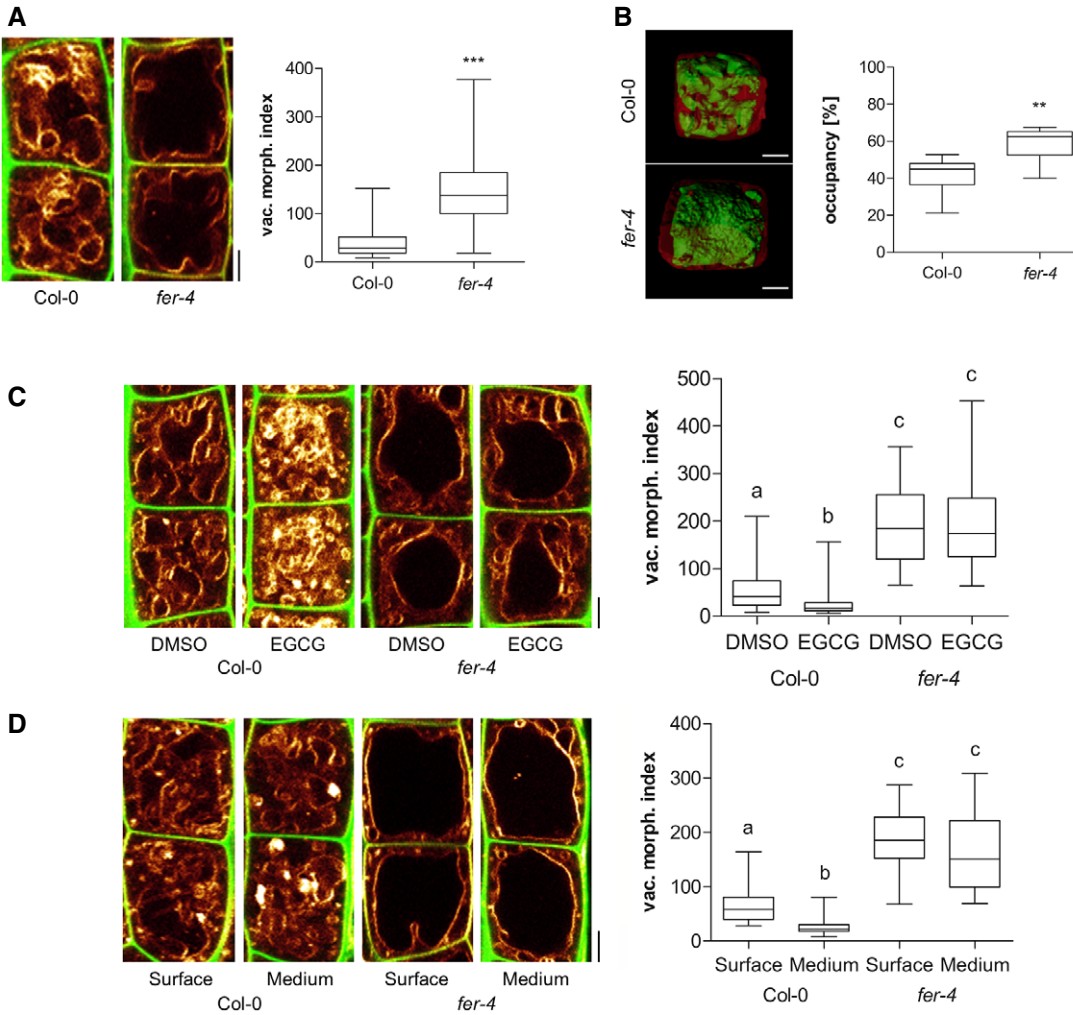

**Figure 4.  Putative cell wall sensor FERONIA impacts on vacuolar size.**

A–D  Representative images and quantification of vacuolar morphology of late meristematic atrichoblast cells. In panels (A, C and D), PI (green) and MDY-64 (yellow) staining depicts cell wall and vacuolar membrane, respectively. (A) Vacuolar morphology of Col-0 (n = 64) and *fer-4* (n = 60). Mann–Whitney U-test (***P < 0.001). (B) 3-D reconstructions of PI-stained cell wall (red) and BCECF-stained vacuole (green). Boxplot depicts vacuolar occupancy of Col-0 control (n = 12) and *fer-4* (n = 10) mutant cells. Mann–Whitney U-test (**P < 0.01). (C) Col-0 (n = 52) and *fer-4* (n = 44) seedlings were treated with solvent DMSO or 50 μM EGCG for 22 h on solid medium. Kruskal–Wallis test followed by Dunn's multiple comparison test (b: P < 0.05, c: P < 0.001). (D) Col-0 (n = 28) and *fer-4* (n = 28) mutant seedling roots were grown on the surface or into the matrix of 2% agar-containing solid medium. Kruskal–Wallis test followed by Dunn's multiple comparison test (b: P < 0.05, c: P < 0.001). In all panels scale bars: 5 μm. Boxplots: Box limits represent 25th percentile and 75th percentile; horizontal line represents median. Whiskers display min. to max. values. Representative experiments are shown.

Source data are available online for this figure.

hypothetical role of LRX in cell wall sensing (Humphrey *et al*, 2007). LRX1 and LRX2 are mainly associated with root hair growth, while LRX8-LRX11 are pollen specific (Baumberger *et al*, 2003a,b). Here, we concentrated on LRX3, LRX4 and LRX5, because they have been suggested to redundantly impact root and shoot growth (Draeger *et al*, 2015). In agreement, *lrx3 lrx4 lrx5* triple mutants displayed a pronounced enlargement of the vacuolar lumina when compared to the wild type (Fig 5A) and the *lrx* single and double mutants (Appendix Fig S5A). Similar to *fer* mutants, these changes also resulted in vacuoles occupying more space in the late meristematic, epidermal cells (Fig 5B). Notably, epidermal cell length was mostly unaffected in *lrx3 lrx4 lrx5* mutant background

(Appendix Fig S5B). *lrx3 lrx4 lrx5* triple mutant vacuoles were, moreover, resistant to EGCG treatments as well as to external constraints by the substrate (Fig 5C and D). In agreement, *lrx3 lrx4 lrx5* mutants displayed increased resistance to the root growth inhibitory effect of EGCG when compared to wild type (Appendix Fig S5C and D) as well as *lrx* single and double mutants (Appendix Fig S5E). We accordingly conclude that extracellular LRX proteins are redundantly involved in setting the intracellular expansion of the vacuole.

We noted that not only the vacuoles, but also the overall plant phenotype, of *lrx3 lrx4 lrx5* triple mutants closely resembled the appearance of *fer* mutants (Fig 6A). Notably, salt stress in the root

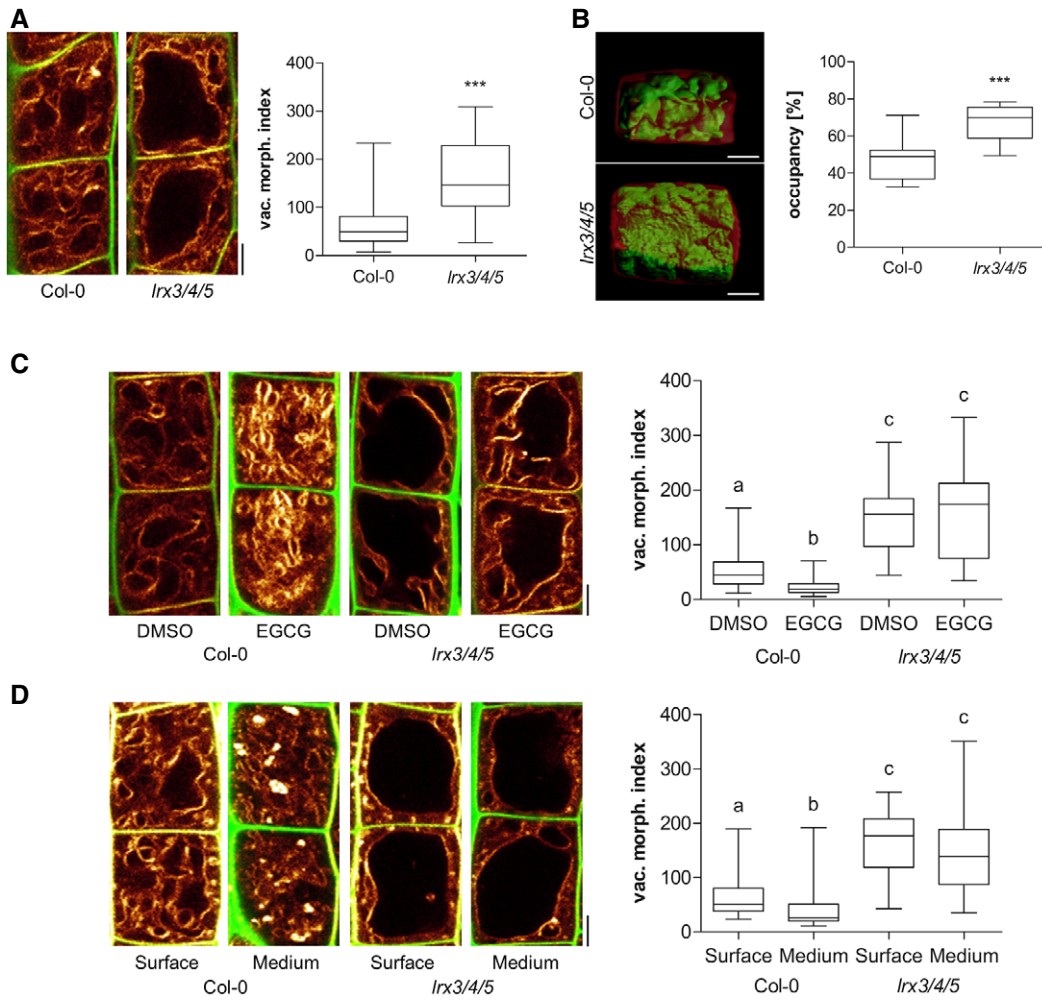

**Figure 5. Extracellular LRX proteins are required to constrain vacuolar expansion.**

A–D  Representative images and quantification of vacuolar morphology of late meristematic atrichoblast cells. In panels (A, C and D), PI (green) and MDY-64 (yellow) staining depicts cell wall and vacuolar membrane, respectively. (A) Vacuolar morphology of Col-0 control (*n* = 52) and *lrx3/4/5* triple mutants (*n* = 48). Mann–Whitney *U*-test (***P < 0.001). (B) 3-D reconstructions of PI-stained cell wall (red) and BCECF-stained vacuole (green) of late meristematic cells. Boxplot depicts vacuolar occupancy of the cell in Col-0 control (*n* = 11) and *lrx3/4/5* (*n* = 10). Student's *t*-test (***P < 0.001). (C) Col-0 (*n* = 40–44) and *lrx3/4/5* (*n* = 36) seedlings were treated with DMSO or 50 μM EGCG for 22 h on solid medium. Kruskal–Wallis test followed by Dunn's multiple comparison test (b: *P* < 0.01, c: *P* < 0.001). (D) Col-0 (*n* = 40–48) and *fer-4* (*n* = 28–32) seedlings were grown on the surface or into the matrix of 2% agar-containing solid medium. Kruskal–Wallis test followed by Dunn's multiple comparison test (b: *P* < 0.05, c: *P* < 0.001). In all panels scale bars: 5 μm. Boxplots: Box limits represent 25th percentile and 75th percentile; horizontal line represents median. Whiskers display min. to max. values. Representative experiments are shown.

Source data are available online for this figure.

has been recently shown to damage, among others, the cell wall. Even though it cannot be ruled out that salt stress also triggers additional defects in the plasma membrane or cytoplasm, it seems that the salt-induced defects in the cell wall are sensed by FER (Feng *et al*, 2018). In agreement with our assumptions, salt sensitivity of *lrx3 lrx4 lrx5* triple mutant largely resembled *fer* single mutants, suggesting that the FER and LRX proteins might function in the same signalling process. In agreement with these assumptions, the morphological and cellular phenotypes of *fer lrx3 lrx4 lrx5* quadruple mutants were not distinguishable from the *fer* single mutants (Fig 6A). Likewise, the root growth response to salt stress was not enhanced in *fer lrx3 lrx4 lrx5* quadruple mutants when compared to *fer* single mutants (Appendix Fig S6). Collectively, this set of data indicates that FER and LRX reside in the same pathway.

Subsequently, we assessed how LRX could function together with FER in extracellular sensing. Other pollen-specific members of the LRX protein family (LRX8-LRX11) have been recently shown to bind to RALF4 and RALF19 via its LRR domains (Mecchia *et al*, 2017). We confirmed such a potential interaction for seedling expressed LRX proteins by co-immunoprecipitating the HA-tagged N-terminal part of LRX4 (LRR4-HA) together with FLAG-tagged RALF1 (Fig 6B).

As FER was also shown to bind RALF1 (Haruta *et al*, 2014), the LRX proteins could be in principle linked to FER-dependent processes via RALF peptide binding. We, hence, tested the contribution of LRX3-5 to the RALF1-sensitive root growth. Root growth of *fer* mutants is strongly resistant to RALF1 application, whereas *lrx3 lrx4 lrx5* triple mutant remained sensitive and showed only slight

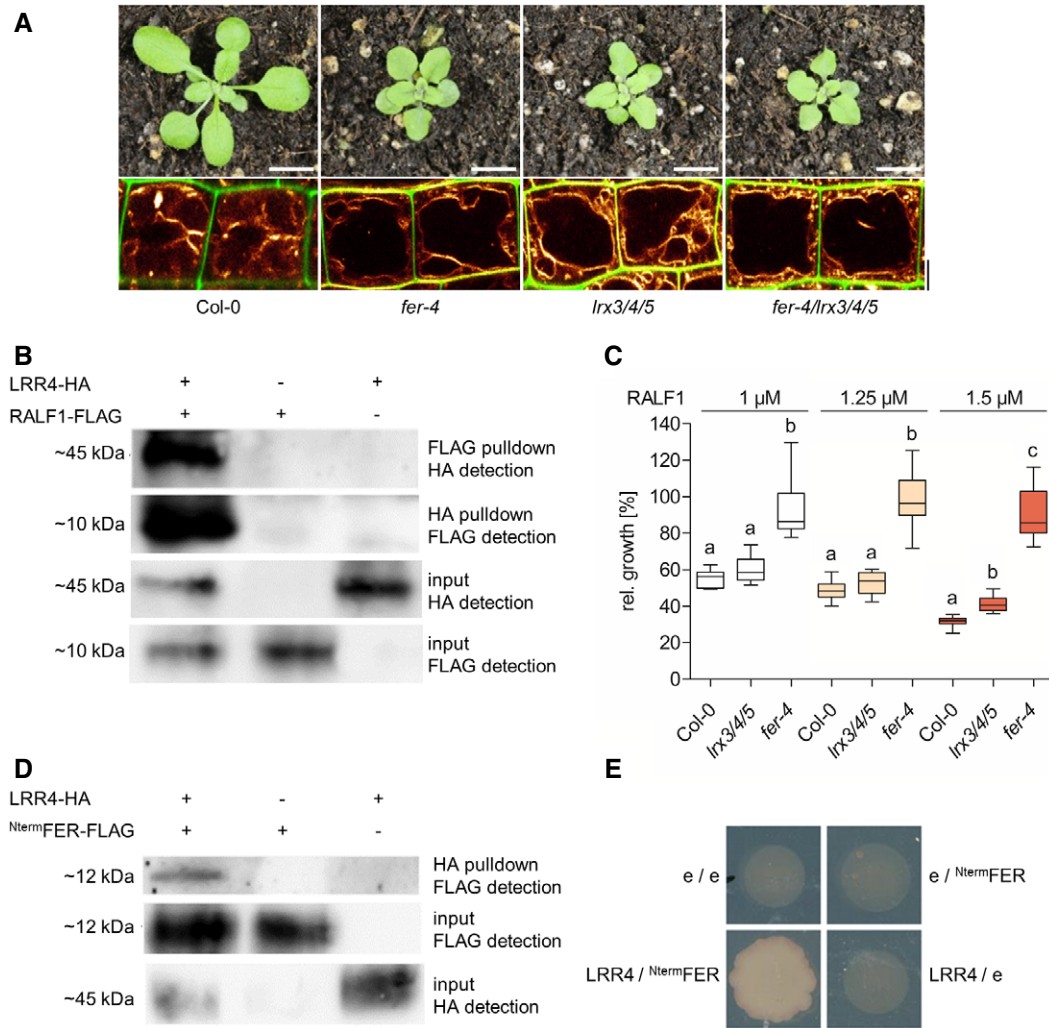

**Figure 6. LRX and FERONIA jointly sense extracellular signals.**

A  Rosette phenotype of 3-week-old Col-0, *fer-4*, *lrx3/4/5* and *fer-4/lrx3/4/5* (upper panel). Vacuolar morphology (MDY-64-stained) of late meristematic atrichoblast cells of Col-0, *fer-4*, *lrx3/4/5* and *fer-4/lrx3/4/5* (lower panel). Scale bars: 1 cm (upper row) and 5 μm (lower row).
B  LRR4-HA and RALF1-FLAG were transiently expressed (as indicated by + and −) in *Nicotiana benthamiana*. Immunoprecipitation and subsequent detection of the proteins by Western blotting were done as labelled.
C  Relative root length [% of control] of Col-0 (n = 11–13), *lrx3/4/5* (n = 9–13) and *fer-4* (n = 11–12) after 3 days of RALF1 treatment. Statistical analyses were performed for each concentration using one-way ANOVA followed by Bonferroni post test (1 and 1.25 μM b: $P < 0.001$; 1.5 μM b: $P < 0.05$, c: $P < 0.001$). Boxplots: Box limits represent 25th percentile and 75th percentile; horizontal line represents median. Whiskers display min. to max. values. Representative experiment is shown.
D  *Nicotiana benthamiana* was transiently transformed with LRR4-HA and NtermFER-FLAG. Immunoprecipitation and subsequent detection of the proteins by Western blotting were done as labelled.
E  Yeast was transformed with either empty plasmids (e/e), NtermFER-FLAG or LRR4-HA with the empty plasmid, or both NtermFER-FLAG and LRR4-HA. Growth of yeast on quadruple drop-out medium was only observed in cells transformed with NtermFER-FLAG and LRR4-HA.

Source data are available online for this figure.

resistance to RALF1 at higher concentrations when compared to a wild-type control (Fig 6C). In addition, the rapid impact of RALF1 on apoplast alkalinisation was completely abolished in *fer* mutants, but still detectable in *lrx3 lrx4 lrx5* triple mutants (Appendix Fig S7). Accordingly, we conclude that root meristem-expressed *LRXs*, such as *LRX3*, *LRX4* and *LRX5*, modulate sensitivity to RALF1, but are not absolutely required for the FER-dependent perception of RALF1.

The *lrx3 lrx4 lrx5* largely resembles *fer* mutant phenotypes, but the comparably weak resistance to RALF1 suggests that LRX may

have additional functions in FER-dependent processes. Interestingly, truncated LRX proteins that lack the EXT cell wall binding domain associate with membranes (Fabrice *et al*, 2018), suggesting that the N-terminus of LRX4 interacts with some membrane component. Therefore, we next assessed whether LRX and FER may reside within a complex. Accordingly, we expressed the LRR4-HA together with a GFP-tagged FER in *Nicotiana benthamiana*. Co-immunoprecipitation of full-length FER and LRR4 indicated an association of these two proteins in a complex (Appendix Fig S8). Strikingly,

LRR4-HA co-immunoprecipitated also with a truncated N-terminal part of FER (NtermFER; Fig 6D), suggesting that a part of the extracellular malectin-like domains is sufficient for the association with LRX4. In addition, the LRR4 domain also interacted in a yeast two-hybrid approach with the NtermFER (Fig 6E), suggesting that FER and LRX4 can directly interact with each other.

Next, we overexpressed a truncated, EXT-lacking version of LRX4 in *Arabidopsis* to further assess the contribution of the EXT domain. Overexpression of the citrine-tagged LRR4 (*p35S::LRR4-citrine*) caused dominant negative phenotypes in *Arabidopsis*, which were morphologically reminiscent to the *lrx3 lrx4 lrx5* triple or *fer* single mutants (Fig 7A). In agreement, the expression of the LRR4 domain was also sufficient to induce bigger luminal vacuoles in late meristematic, epidermal cells (Fig 7B). Moreover, LRR4 overexpressors were insensitive to EGCG-induced reduction in vacuolar size (Fig 7B). This set of data suggests that the EXT cell wall binding domain is critically important for the LRX/FER-dependent cell wall sensing mechanism.

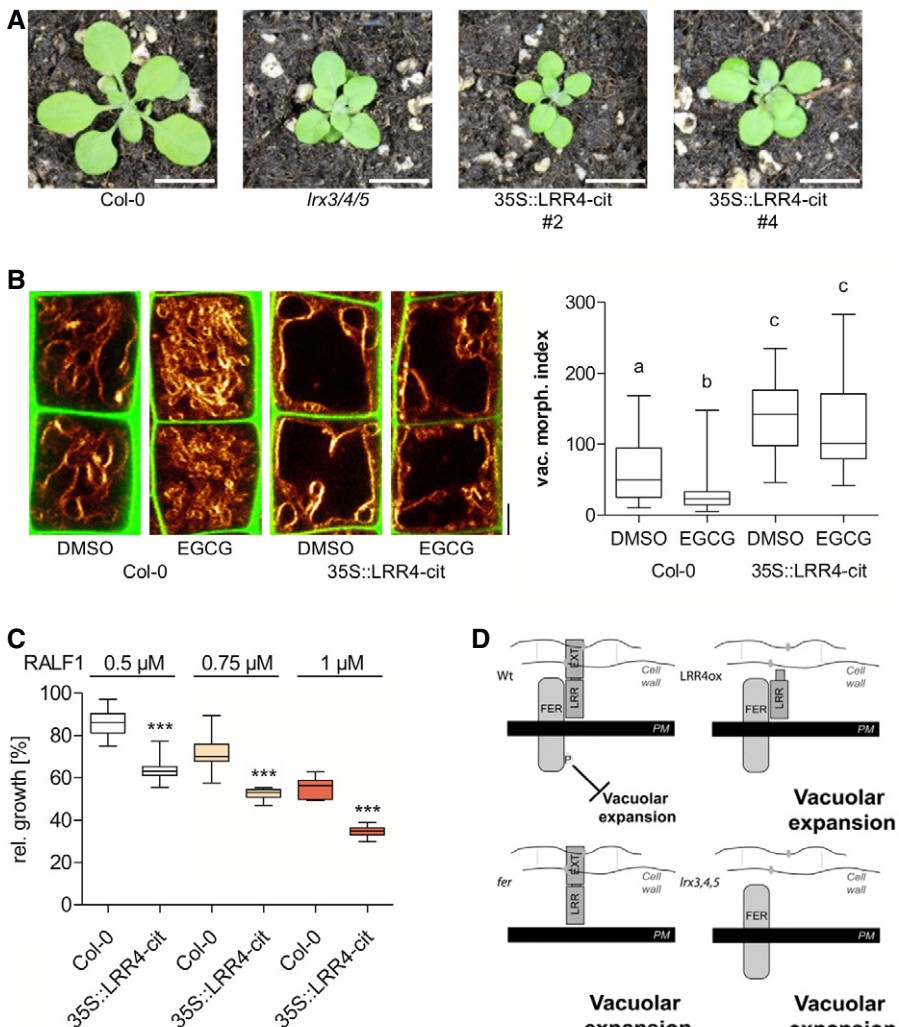

**Figure 7. EXT-domain of LRXs is crucial for cell wall sensing.**

A  Rosette phenotype of 3-week-old Col-0, *lrx3/4/5* and *35S::LRR4-cit*.

B  Representative images and quantification of vacuolar morphology of late meristematic cells of Col-0 and 35S::LRR4-cit. PI (green) and MDY-64 (yellow) staining depict cell wall and vacuolar membrane, respectively. Col-0 (*n* = 36) and *35S::LRR4-cit* (*n* = 32–36) seedlings were treated with DMSO or 50 µM EGCG for 22 h on solid medium. Kruskal–Wallis test followed by Dunn's multiple comparison test (b: *P* < 0.05, c: *P* < 0.001).

C  Relative root length of Col-0 (*n* = 11–13) and *35S::LRR4-cit* (*n* = 11–14) after 3 days of RALF1 treatment. Statistical analyses were performed for each concentration using Student's *t*-test (***P* < 0.001).

D  Schematic model depicting the FER- and LRX-dependent cell wall sensing mechanism, which limits vacuolar expansion (WT) in late meristematic cells. In contrast, dysfunctional cell wall perception in *lrx3 lrx4 lrx5*, *fer-4* and *35S::LRR4-cit* triggers premature vacuolar expansion.

Data information: Scale bars: 1 cm (A) and 5 µm (B). Boxplots: Boxplots: Box limits represent 25th percentile and 75th percentile; horizontal line represents median. Whiskers display min. to max. values. Representative experiments are shown. Representative experiments are shown.

Source data are available online for this figure.

The expression of LRR4 largely phenocopied the *lrx3 lrx4 lrx5* triple mutant, but, intriguingly, the *p35S::LRR4-citrine* expressing roots were not slightly resistant (as seen for the triple mutant), but hypersensitive to RALF1 treatments when compared to its respective wild-type control (Fig 7C). This finding illustrates again that LRXs indeed seem to modulate RALF sensitivity, but also strongly suggests that the function of LRX in FER-dependent cell wall sensing is at least partially distinct from its impact on FER-dependent perception of the RALF1 peptide.

## Discussion

Vacuoles are essential for plant development (Rojo *et al*, 2001), and interference with vacuolar function evokes severe cell expansion and developmental defects (Schumacher *et al*, 1999; Li *et al*, 2005). The size of vacuoles, moreover, correlates with individual cell size in plant cell cultures (Owens & Poole, 1979) as well as in the root tip of *Arabidopsis* (Berger *et al*, 1998; Löfke *et al*, 2013), proposing a role in cell size determination (Löfke *et al*, 2015). Here, we further support these previous assumptions and show that the increase in vacuolar volume allows for rapid cellular elongation with relatively little *de novo* production of cytosolic content. Accordingly, we define the vacuolar size as a suitable parameter to quantify an important intracellular process, marking the regulation of cell expansion. Using this read-out, our data suggest that a FER-dependent cell wall sensing mechanism impacts on intracellular processes, including the expansion of the vacuole. FER has been previously proposed to be required for cell wall sensing (Shih *et al*, 2014), but molecular interactors involved in the underlying mechanism were largely unknown. We used the vacuolar size as a quantitative, cellular trait to study FER-dependent mechanisms and show that interaction of extracellular LRX with FER contributes to extracellular sensing (Fig 7D). We propose that this LRX- and FER-dependent module integrates the cell wall status with intracellular growth processes, such as the expansion of the vacuole. It remains to be seen precisely how LRX/FER signalling at the cell surface leads to the modulation of vacuolar size. FER-dependent phosphorylation acts as a signalling relay (Shih *et al*, 2014; Du *et al*, 2016; Haruta *et al*, 2018) and modulates, among others, the Rho of plants (ROP) guanine nucleotide exchange factor 1 (GEF1; Duan *et al*, 2010). ROP GEF1 in turn activates RAC/ROP GTPases (Rho-related molecular switches in plants; Duan *et al*, 2010), which are known regulators of actin dynamics. Notably, the actin cytoskeleton surrounds the vacuole and contributes to the regulation of vacuolar size (Scheuring *et al*, 2016). This hypothetical link could in principle connect the LRX/FER-dependent, extracellular sensing mechanism with the intracellular control of vacuolar expansion.

Our work proposes LRX as a physical link between the plasma membrane-localised FER on one side (via the LRR domain) and the cell wall (via the EXT domain) on the other side (Fig 7D). It is currently unknown which cell wall component is recognised by the EXT domain. The PME inhibitor EGCG induces smaller vacuoles in a FER- and LRX-dependent manner. Accordingly, it is tempting to speculate that pectin may play also a role in this LRX/FER-dependent process.

On the other hand, not only FER, but also THE1, binds to RALF peptides, proposing conserved interaction partners for

CrRLK1Ls (Gonneau *et al*, 2018). It is similarly likely that LRX proteins also interact with a variety, if not all, *Cr*RLK1Ls. It remains, however, unclear precisely how FER and other *Cr*RLK1Ls interact with RALFs and/or LRX proteins. Future studies are required to elucidate whether they form a trimeric complex or rather binding of two proteins excludes interaction with the third. Moreover, recent work on salt-induced cell wall damages proposes that FER binds also to polygalacturonic acid, the backbone of pectin (Feng *et al*, 2018). Whereas animal and bacterial malectin domains indeed display carbohydrate-interaction surfaces, recent work questions that the malectin domains of plant *Cr*RLK1L directly bind to oligosaccharides (Moussu *et al*, 2018). Instead, the crystal structure of the tandem malectin domains of *Cr*RLK1Ls proposes a protein–protein interface (Moussu *et al*, 2018). Based on our work, we assume that LRXs binds via the LRR domain to FER and bridge FER via the EXT domain on its other side with cell wall components, such as pectin. However, further research is needed to assess this assumption and to reveal how a presumable binding of the EXT domain to pectin (or other cell wall components) may impact on the interaction of LRX and FER.

Both LRX and FER proteins are essential for constraining the vacuole, suggesting that the LRX-FER interaction is required for its signalling to the underlying cell. FER has a RALF-dependent scaffold function during immune responses (Stegmann *et al*, 2017) and pollen expressed LRX function in RALF4/19-dependent regulation of pollen tube's cell wall integrity (Mecchia *et al*, 2017). Our data reveal that *LRR4* overexpression largely resembles the morphological and cellular phenotype of *lrx3 lrx4 lrx5* mutants. In contrast, the *lrx* triple mutant and *LRR4* overexpression are partially resistant and hypersensitive to RALF1, respectively. This finding proposes that the roles of LRX in FER-dependent cell wall and RALF sensing are at least partially distinct. Accordingly, we conclude that LRX and FER form distinct complexes, allowing them to integrate very diverse processes.

Here, we propose that the LRX/FER-dependent feedback mechanism aligns the cell wall status with the intracellular expansion of the vacuole. This mechanism, consequently, ensures the spatial and temporal coordination of cell wall acidification/loosening with the increase in vacuolar size, which ultimately effects cytosol homeostasis during rapid cell expansion.

## Materials and Methods

### Plant material and growth conditions

Most experiments were carried out in *A. thaliana* (Col-0 ecotype). The following plant lines were described previously: *fer-4* (Shih *et al*, 2014), *fer-2* (Deslauriers & Larsen, 2010), *lrx3/lrx4/lrx5* (Draeger *et al*, 2015), *pUBQ10::VAMP711-YFP* (Geldner *et al*, 2009), *pER8::GFP-SAUR19* (Spartz *et al*, 2014), pHusion (Gjetting *et al*, 2012), *pFER::FERKR-GFP* (in *fer-4*; Shih *et al*, 2014), pFER::FER-GFP (in *fer-4*; Shih *et al*, 2014) and *35S::LRR4-citrine* (Fabrice *et al*, 2018). Seeds were stratified at 4°C for 2 days in the dark and were grown on vertically orientated ½ Murashige and Skoog (MS) medium plates containing 1% sucrose under a long-day regime (16-h light/8-h dark) at 20–22°C.

### Chemicals and RALF1

All chemicals were dissolved in dimethyl sulfoxide (DMSO) and applied in solid or liquid ½ MS medium. MDY-64 was obtained from Life Technologies (CA, USA), β-estradiol, propidium iodide (PI), 2′,7′-Bis(2-carboxyethyl)-5(6)-carboxyfluorescein acetoxymethyl ester (BCECF-AM) from Sigma (MO, USA), and fusicoccin (FC) and epigallocatechin gallate (EGCG) from Cayman Chemical (MI, USA). The RALF1 peptide (mature RALF1, amino acid sequence: ATTKYISYQSLKRNSVPCSRRGASYYNCQNGAQANPYSRG CSKIARCRS) was obtained from Peptron (KOR).

### Phenotype analysis

Vacuolar morphology index and occupancy were quantified in 6-day-old seedlings. Confocal images were analysed using ImageJ (vacuolar morphology index) or processed using Imaris (vacuolar occupancy of cells). To calculate the vacuolar morphology index, the longest and widest distance of the biggest luminal structure was measured and multiplied (Löfke *et al*, 2015). The atrichoblast cells were quantified before the onset of elongation (late meristematic). To depict this region, the first cell being twice as long as wide was considered as the onset of elongation. Starting from this cell, the next cell towards the meristem was excluded (as it usually shows either partial elongation and/or already substantial vacuolar expansion), and vacuoles of the subsequent 4 cells were quantified as described previously (Scheuring *et al*, 2016). For the analysis of occupancy, 1 cell in this region was used. Vacuolar shape/size was quantified in at least 8 roots (unless stated otherwise in the figure legend). MDY-64 and BCECF staining was performed as described previously (Scheuring *et al*, 2015). Plant rosette phenotype evaluation was performed 3 weeks after germination.

For the analysis of root growth under salt stress conditions, 5-day-old seedlings were transferred to ½ MS plates supplemented with 100 mM NaCl and grown for another 9 days. Plates were scanned and root length assessed using ImageJ.

### Cell length analysis

Six-day-old seedlings were used to quantify atrichoblast cell length. The region of analysis was similar to the quantification of vacuolar morphology index (see above). The distance covering four cells was measured per root and divided by four, resulting in the average cell length per root.

### RALF1 and EGCG root length assay

Three-day-old seedlings (*n* = 10–12) were transferred for another 3 days to 3 ml liquid ½ MS medium containing 1 μM RALF1 or 25 μM EGCG or the appropriate amount of solvent (water or DMSO, respectively). The seedlings were then placed on solid MS plates prior to scanning. Root length was analysed using ImageJ.

### Apoplastic pH visualisation in root cells

Five-day-old seedlings were treated for 10 min in liquid ½ MS medium supplemented with 1 μM RALF1 or the appropriate amount of solvent (water) or 6-day-old seedlings were treated in liquid ½ MS medium supplemented with 5 μM FC or DMSO for the indicated time. Subsequently, the seedlings were transferred to a block of solid ½ MS medium containing 1 mM of 8-hydroxypyrene-1,3,6-trisulfonic acid trisodium salt (HPTS; Sigma-Aldrich). This block was then mounted on a microscope slide and instantly used for imaging. Image processing was performed as described in Barbez *et al* (2017). Four transversal cell walls before the onset of elongation were quantified per root, and these values were averaged per root.

### Agar penetration assay

Sixty millilitres of ½ MS medium containing 2% of plant agar was poured into square Petri dishes (12 × 12 cm). Approximately 2 cm of medium at the top of the plate was removed with a scalpel, and small notches were generated on the surface using a toothpick (enabling root penetration into the medium). Subsequently, a single seed was placed in each notch.

### 3-D reconstruction of vacuoles

Imaris 8.4.0 was used for the reconstruction of cell and vacuole volumes. Based on the PI channel, every 3rd slice of the z-stack was utilised to define the cell borders using the isoline, magic wand or manual (distance) drawing functions in the manual surface creation tool. After creating the surface corresponding to the entire cell, a masked channel (based on BCECF) was generated by setting the voxels outside the surface to 0. Subsequently, a second surface (based on the masked BCECF channel) was generated automatically with the smooth option checked. The obtained surface was visually compared to the underlying BCECF channel, and, if necessary, the surface was fitted to the underlying signal by adjusting the absolute intensity threshold slider. Finally, volumes of both surfaces were extracted from the statistics window.

### Confocal microscopy

For image acquisition, a Leica TCS SP5 (DM6000 CS) confocal laser-scanning microscope, equipped with a Leica HCX PL APO CS 63 × 1.20 water-immersion objective, was used. MDY-64 was excited at 458 nm (fluorescence emission: 465–550 nm), GFP and BCECF at 488 nm (fluorescence emission: 500–550 nm), YFP at 514 nm (fluorescence emission: 525–578 nm) and PI at 561 nm (fluorescence emission: 644–753 nm). Roots were mounted in PI solution (0.02 mg/ml) to counterstain cell walls. Z-stacks were recorded with a step size of 420 nm. On average, 36 slices in z-direction were captured, resulting in an average thickness of approximately 15 μm. The argon laser power was set to 30%, the AOTF for the 488 nm laser line was set to 2%, and the HyD gain was set to 300 (BCECF channel). The AOTF for the 561 nm laser line was set to 20%, and the PMT gain was set to 900 (PI channel). The pinhole was set to 111.6 μm. HPTS was excited at 405 nm (protonated form) with 4% laser line intensity and at 458 nm (deprotonated version) with the AOTF set to 100%, and the argon laser power was set to 60%. The gain was set to 799. HPTS images were acquired in sequential scan mode with the detection window set to 499–546 nm.

## Co-immunoprecipitation assay

For pulldown and co-IP analysis of FER and LRR4, Agrobacteria containing *pFER::FER-GFP* (Escobar-Restrepo *et al*, 2007) and/or *p35S::LRR4-HA* (Fabrice *et al*, 2018) were infiltrated into *Nicotiana benthamiana* leaves. After 48 h of infiltration, the tobacco leaves were excised and grinded in liquid nitrogen. The tissue powder was re-suspended in extraction buffer [50 mM HEPES-KOH (pH 7.6), 150 mM NaCl, 1 mM DTT, 1 mM PMSF, protease inhibitor and 1% NP-40]. The suspension was incubated on ice for 30 min and then centrifuged at 17,000 *g* for 20 min at 4°C. The supernatant obtained was then incubated with GFP-trap agarose beads or anti-HA agarose beads for 3–4 h at 4°C on a rotating shaker. After incubation, the beads were washed three times with the extraction buffer containing 0.1% NP-40 and boiled in SDS–PAGE loading buffer for 15 min at 75°C. The immunoprecipitates were then run on a 10% SDS–PAGE and transferred to nitrocellulose membrane to perform Western blotting.

## RALF1 and ^NtermFER construct

For *RALF1_2FLAG* overexpression, the full-length coding sequence was amplified using the primers RALF1oE_F GGTACCATGGA CAAGTCCTTTACTC and RALF1oE_R CTGCAGAACTCCTGCAAC GAGCA. The fragment was cloned into pJET1.2 (Thermo Scientific). A correct clone was cut with *Pst*I and *Xba*I and fused with a *Pst*I-2FLAG-stop *Xba*I fragment. The resulting *RALF1_2FLAG* was cut with *Kpn*I and *Xba*I and cloned into pART7 vector (Gleave, 1992), using the same enzymes. The resulting *35S:RALF1_2FLAG* construct was cut out by *Not*I and cloned into the binary vector pBART (Stintzi & Browse, 2000).

For overexpression of the ^NtermFER extracellular domain, the coding sequence was amplified with the primers FER_ECD_F CT CGAGATGAAGATCACAGAGGGAC and FER_ECD_R CTGCAGGC CGTCTGAGAAGCACTG, cloned into pJET1.2 (Thermo Scientific). A correct clone was cut with *Xho*I as well as *Pst*I and cloned into pART7 (Gleave, 1992) containing a 2FLAG coding sequence with a *Pst*I site at the 5′ end, cut with *Xho*I and *Pst*I.

## Yeast two hybrid

For the yeast two-hybrid experiment, the coding sequence of the LRR domain of LRX4 was amplified using the primers y2hs_LRR4_F GGATCCAAGCTCTTGATAACCGGAAG and y2h_LRR4_R CTCGA GCTATCCACAATCCACCGAAGGCCG and cloned into pJET1.2 (Thermo Scientific). A correct clone was cut with *Bam*HI and *Xho*I and cloned into pGBKT7 cut with *Bam*HI and *Sal*I. The extracellular domain coding sequence of FER (^NtermFER) was amplified using the primers Y2H_FER_F CATGAATTCCGTATATGGATCTCCGAT and Y2H_FER_R ATGCCCGGGTCCGCCGTCTGAGAAGCAC and cloned into pJET1.2 (Thermo Scientific). A correct clone was cut with *Eco* R I and *Xma*I and cloned into pGADT7 cut with *Eco* R I and *Xma*I. Transformation of the yeast strain PJ69-4A (James *et al*, 1996) was done following standard procedures and quadruple drop-out medium lacking Leu, Trp, His and Ade were used to screen for positive interactions.

**Expanded View** for this article is available online.

## Acknowledgements

We are grateful to Y. Belkhadir, A.T. Fuglsang, N. Geldner, B. Gray, P.B. Larsen, C. Luschnig, G. Monshausen and C. Zipfel for sharing published material; Elke Barbez for critical reading of the manuscript; Jit Thacker for help with preparing the manuscript; and the BOKU-VIBT Imaging Centre for access and expertise. This work was supported by the Austrian Academy of Sciences (ÖAW) (DOC fellowship to K.D.), Vienna Research Group (VRG) programme of the Vienna Science and Technology Fund (WWTF), the Austrian Science Fund (FWF) (Projects: P26568-B16 and P26591-B16), the European Research Council (ERC) (Starting Grant 639478-AuxinER) (to J.K-V.), and the Swiss National Science Foundation (SNF) (31003A_166577/1, to C.R.).

## Author contributions

KD, CR and JK-V designed research. KD conducted most experiments; SG, AH and CR performed pulldown and yeast experiments; MIF established multiple mutants. KD, SG, AH, MIF, CR and JK-V analysed data. KD and JK-V wrote and all co-authors commented on the manuscript.

## Conflict of interest

The authors declare that they have no conflict of interest.

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
