## [Review Process File · The EMBO Journal]

Extracellular matrix sensing by FERONIA and Leucine-Rich Repeat Extensins controls vacuolar expansion during cellular elongation in *Arabidopsis thaliana*.

Kai Dünser, Shibu Gupta, Aline Herger, Mugurel Feraru, Christoph Ringli and Jürgen Kleine-Vehn.

Review timeline:

Submission date:	25 th July 2018
Editorial Decision:	11 th September 2018
Revision received:	26 th November 2018
Editorial Decision:	20 th December 2018
Revision received:	10 th January 2019
Accepted:	21 st January 2019

Editor: Elisabetta Argenzio

Transaction Report:

1st Editorial Decision

11th September 2018

Thank you for submitting your manuscript on the modulation of cellular elongation in *Arabidopsis thaliana* root to The EMBO Journal. Your study has been sent to three referees for evaluation, and we have now received reports from them, which are enclosed below for your information.

As you can see, the referees concur with us on the overall interest of your findings. However, they also raise some points that need to be addressed before they can support publication in The EMBO Journal. In particular, referees #1 and #2 point out that the discussion should be expanded to integrate the presented findings in the context of the existing literature. Referee #3 asks you to quantify the cell size in wild-type, fer and lrx mutants (points 3 and 4).

Addressing these issues as suggested by the referees is required to warrant publication in The EMBO Journal. Given the overall interest of your study, I would like to invite you to revise the manuscript in response to the referee reports.

REFeree REPORTS

Referee #1:

The manuscript by Dünser et al. investigates modulation of cellular elongation in the *Arabidopsis* root via signaling integration of cell wall properties with intracellular changes. To do so, the authors first establish vacuole size as a readout to quantitate intracellular growth processes. This methodology enabled subsequent testing of a FERONIA-dependent module's ability to sense changes in cell wall stiffness and provide feedback for vacuolar morphogenesis. A role for FER in cell wall sensing has previously been proposed, though additional players in the underlying mechanism were unknown. A major outstanding question has been the manner of interaction between FER and the cell wall. To this end, the manuscript raises several points of particular

interest:

- 1) LRX3, 4, and 5 function in the same pathway with FER and participate in perception of cell wall properties, specifically loosened or stiffened cell walls.
- 2) The LRX/FER module integrates cell wall status with intracellular changes, as indicated by changes in vacuole size.
- 3) LRX/FER may function in a complex and the EXT cell wall binding domain of LRX4 is required for perception of cell wall properties.

FER and other related RLKs are of considerable interest to the research community as suspected cell wall sensors. The characterization of a novel role of LRX proteins in the context of FER signaling is certainly an important step toward elucidating the mechanism by which FER facilitates extra- and intracellular communication. Here, the authors propose that LRXs may act as a physical bridge between plasma membrane-localized FER and the cell wall. I agree that this idea is initially supported by evidence that LRX4 physically interacts with FER and its EXT cell wall binding domain is important for sensing extracellular signals. The revelation of LRX function in perception of cell wall properties as at least partially distinct from its participation in RALF perception by FER provides insight into the potential for multiple mechanisms of LRX-mediated extracellular sensing. Thorough establishment of a method to assay intracellular responses via vacuole size is useful to the broader research community.

I believe that conclusions are drawn with care and supported by the data shown. I have two suggestions to strengthen the manuscript:

- 1) Most previous information for RLK-mediated cell wall sensing is available for THESEUS1. A brief mention of how results from FER relate to what is known about THESEUS1 will integrate information gained from this study with broader RLK research.
- 2) Lines 177-179: From the co-IP presented in Fig 5d, the authors suggest that part of the extracellular malectin-like domains is sufficient for FER interaction with LRX4. Could the authors include a brief discussion on this result in light of previous hypotheses (summarized by Li, Wu, and Cheung, 2016) that the malectin domains of FER and related RLKs themselves may interact with carbohydrates in the cell wall and contribute to sensing.

Additional minor comments:

- 3) Line 121: no section F in Fig 3
- 4) Line 132 - 138: Co-expression data with FER is only shown for LRX3. Is expression data similar for LRX4 and LRX5? Are these the only three LRXs with spatial expression of interest (i.e., expressed in root)?
- 5) Since the triple mutant *lrx3 lrx4 lrx5* is used for the cell wall sensing experiments, I assume the three genes are likely functionally redundant? In the discussion, the authors may wish to briefly comment on this redundancy and what it might mean for function in FER-dependent cell wall sensing.

The manuscript is well written with questions, hypotheses, and data clearly presented.

Referee #2:

This manuscript by Dunser et al. reports the measurements of vacuole size/morphology in Arabidopsis root cells to demonstrate some genetic components required for maintaining the normal vacuole development. They have chosen to examine whether genetic mutation or overexpression of a receptor like kinase FERONIA, a family of potential signaling proteins localized in extracellular side of the plasma membrane, leucine-rich-repeat (LRR) extensin, and auxin responsive signaling protein, SAUR19 as those proteins were previously established to be some of the perhaps many critical components of cell growth by other laboratories. In this study, the authors have developed

3D imaging techniques with the use of organelle-specific fluorescent dyes in order to visualize the vacuole, quantify vacuolar phenotypes, and compare wildtype and those mutants.

They also examined how various compounds affect vacuolar size by treating Arabidopsis roots with several reagents including a PM proton pump stimulator, fusicoccin, a growth inhibitory peptide, RALF, and a pectin methylesterase inhibitor, EGCG. From those experiments, they conclude that FERONIA as well as LRR-extensin that were previously known to be involved in cell expansion process are also involved in determining vacuole size/morphology. Overall they provide interesting observations on the correlation of FER and LRR-extensin mutations with vacuole morphology. I have various comments to this manuscript which the authors may consider, to improve the manuscript before publication to the EMBO J.

Whereas it is nice that they can find the correlation of vacuole size and the predicted role of PM signaling protein FER and LRR-extensin, in cell expansion, I (probably a majority of readers) would like to see some models for the connection of the cell surface events and vacuole. All the datasets they show are correlative responses under growth-altering physiological or genetic conditions. There are some known mutants involved in vacuole and cell growth (such as V-type H-ATPase, Genes Dev. 1999 Dec 15; 13(24): 3259-3270., The Plant Cell, Vol. 26: 3416-3434 or the original paper which described H⁺ pyrophosphatase defective phenotype). They could mention some mutant phenotypes describing defective function in the vacuole and cell growth in the Discussion section if their experimental results are not available. Clearly there are going to be MANY movements of various organic and inorganic ions between the extracellular fluid, the cytoplasmic fluid and the vacuolar fluid and thus these three compartments are going to see many fluxes in all directions all contributing to the longer term morphological changes they are reporting for the vacuole. Time courses for things like FC and RALF, that act in seconds and a few minutes, rather than hours, would be a tremendous improvement.

Line 145, they mention that salt stress damage the cell wall, but no mention about negative effects of high salts on the plasma membrane nor cytoplasm. Some Salt Overly Sensitive mutants are defective in the plasma membrane transporter or localized at the plasma membrane. They may not assume the negative effect of high salts are only at the cell wall.

Line 212, they state FER-kinase function is required for FER phosphorylation events and cite Ref. 24. In fact, Ref 24 did not demonstrate the kinase dependent role of FER. The three papers that show kinase-dependent role of FER are "Receptor kinase complex transmits RALF peptide signal to inhibit root growth in Arabidopsis, PNAS", "The Receptor-like Kinase FERONIA Is Required for Mechanical Signal Transduction in Arabidopsis Seedlings, Current Biology", and "Comparison of the effects of a kinase-dead mutation of FERONIA on ovule fertilization and root growth of Arabidopsis, FEBS Letters". Those papers should be referred in line 212 instead of Ref.24.

In Supplemental Fig. 7, it would make sense to include the results of LRR4 overexpressing line with Col-0, fer-4, and lrx3/4/5.

They have done great 3D imaging and surface rendering of the vacuole in various different root cell types shown in Fig.1. To me they should be using this 3D assay for all the rest of image analyses shown in Fig. 2, 3, 4 and 6. In the images shown in Fig. 2, 3, 4 and 6 are captured in 2D, which potentially induce bias when different planes of the tissues are scanned, unless they provide what the z depth in the roots these planes are examined.

In the method section of 3D reconstruction of vacuoles, they provide very little information as to the imaging method. What is the pinhole value (either airy unit or absolute value, i.e., nm or micron)? How many optical slices were captured for z stack? They mention step size but no mentioning the number of slices. This number would be useful so that one can get some ideas on the thickness/volume of their images. Please provide either laser power transmission % value of excitation and/or gain setting of confocal imaging. Those would tell us what levels of fluorescent signals were obtained from their specimen.

Fig. 2 figure caption heading indicates "with cell wall acidification". But there are no data of pH measurements in this figure. Please correct this.

Supplemental Fig 4. There is gene expression profile of THE1. No description of this figure in the manuscript text. Either remove the chart or please describe it.

They treated Arabidopsis roots with fusicoccin, acidic pH, pectin methylesterase inhibitor EGCG or induced SAUR19 by inducible expression using estradiol, to examine the effects of those physiological conditions. The timings of incubation with the conditions vary from 2.5 hours to 22 hours. What we see are the end point result. Ideally it should have done with time-course or real-time measurement (see earlier comment). If possible provide some RAPID time course results, or I would suggest to add some cautionary notes on those endpoint measurements in relation to whether the observations are direct or indirect responses.

They propose cell wall sensing and vacuolar morphology/size in the context of cell expansion. In all the images in Fig.2, 3, 4, and 6, they conclude vacuole morphology is affected by genetic mutations or chemical treatment. But these cells do not appear to be affected in terms of cell length or width. Are they proposing that vacuole morphology changes before cells expand? Though incubation time is rather long (hours); does one expect to see some changes in cell length or width?

The pulldown experiments are fairly meaningless since these have been done already in several publications and more important, since the proteins contain hydrophobic transmembrane domains and require detergents to be released from the membrane into various types of macromolecular complexes without proper controls (both positive and negative) it is impossible to evaluate the importance of these results.

Referee #3:

1. It is known from classical studies on plant cytology that plant cells expand through increasing the size of the vacuole as the cell surface - membrane and wall - increases in area. Here the authors describe the change in vacuolar size in root epidermal cells (atrachoblasts) during expansion. They show that the vacuole occupies approximately 98% of the volume of the cell by the end of elongation and that the cytoplasm does not increase appreciably in volume.
2. It is known that cell wall acidification in growing cells can increase cell elongation. The authors describe how three different treatments known to cause changes in cell elongation also resulted in a change in vacuole morphology and the relative volume occupied by the vacuole (vacuolar occupancy). As cells elongate we would expect vacuolar occupancy to increase because all growth is due to increase in vacuolar volume while cytoplasmic volume is more or less constant. Similarly, treatments that reduce cell size would result in a lower vacuolar occupancy volume. This is exactly what the authors show, which is good. This is a nice introductory/confirmatory piece of data.
3. FER has been suggested to be involved in cell wall mechanical sensing. Vacuolar volume of fer is different from wild type. The authors suggest that this indicates that FER regulates vacuolar volume. How do they know that the effect of the fer mutation does not cause a defect in cell expansion/cell size which in turn impacts vacuolar volume? To make their point they should include measures of cell size in wild type and fer mutants. If their assertion is correct, then wild type and fer atrachoblast cells must be the same size. These data are not in the paper at the moment.
4. LRX proteins are required for cell growth and thought to play structural roles in the cell walls. The lrx triple mutant develops defective vacuoles compared to wild type. It would be good if the authors should include the measure of lrx trip mutant atrachoblasts to ensure that the effect that is observed is not a result of the differences in cell size between the different genetic backgrounds.
5. The authors demonstrate that LRX and FER proteins interact.

Major points:

Cell sizes need to be included as outlined under points 3 and 4 above to ensure that the changes on vacuolar morphology are not an indirect effect of cell size. I assume that they authors have these data so this should not be a big deal.

Minor points:

The writing style is poor in places. It would be good if the authors could get editorial comments from someone outside the field. Listed here are some suggested changes, but there are many more.

Line 77; define what you mean by vascular occupancy in the Results section.
Line 104-105; This suggests that the effect is causative. it could be a secondary effect. the authors should change their wording
Line 112: The authors should demonstrate or mention that EGCG-treatment resulted in less elongation in their hands.
Line 120; fer mutants showed enlarged vacuoles compared to what. The authors should state this in the sentence.
Line 136; What does interesting mean?
Line 139 than in wild type.
Line 170; Could it be that FER has a structural role like LRX? These data are consistent with that hypothesis. They provide no support for a signaling role of FER.
Line 183: delete "remarkably". It makes it look as though the authors are trying to oversell their data/results.

1st Revision - authors' response

26th November 2018

Reviewer comments (Dünser et al):

Referee #1:

The manuscript by Dünser et al. investigates modulation of cellular elongation in the Arabidopsis root via signaling integration of cell wall properties with intracellular changes. To do so, the authors first establish vacuole size as a readout to quantitate intracellular growth processes. This methodology enabled subsequent testing of a FERONIA-dependent module's ability to sense changes in cell wall stiffness and provide feedback for vacuolar morphogenesis. A role for FER in cell wall sensing has previously been proposed, though additional players in the underlying mechanism were unknown. A major outstanding question has been the manner of interaction between FER and the cell wall. To this end, the manuscript raises several points of particular interest:

- 1) LRX3, 4, and 5 function in the same pathway with FER and participate in perception of cell wall properties, specifically loosened or stiffened cell walls.
- 2) The LRX/FER module integrates cell wall status with intracellular changes, as indicated by changes in vacuole size.
- 3) LRX/FER may function in a complex and the EXT cell wall binding domain of LRX4 is required for perception of cell wall properties.

FER and other related RLKs are of considerable interest to the research community as suspected cell wall sensors. The characterization of a novel role of LRX proteins in the context of FER signaling is certainly an important step toward elucidating the mechanism by which FER facilitates extra- and intracellular communication. Here, the authors propose that LRXs may act as a physical bridge between plasma membrane-localized FER and the cell wall. I agree that this idea is initially supported by evidence that LRX4 physically interacts with FER and its EXT cell wall binding domain is important for sensing extracellular signals. The revelation of LRX function in perception of cell wall properties as at least partially distinct from its participation in RALF perception by FER provides insight into the potential for multiple mechanisms of LRX-mediated extracellular sensing. Thorough establishment of a method to assay intracellular responses via vacuole size is useful to the broader research community.

I believe that conclusions are drawn with care and supported by the data shown. I have two suggestions to strengthen the manuscript:

> We thank you for your encouraging words and for your constructive comments and suggestions.

1) Most previous information for RLK-mediated cell wall sensing is available for THESEUS1. A brief mention of how results from FER relate to what is known about THESEUS1 will integrate information gained from this study with broader RLK research.

> Thank you for your suggestion to relate our work to a broader RLK perspective. We briefly introduced THE1 in our introductory paragraph and added further integrative comments into the discussion section.

2) Lines 177-179: From the co-IP presented in Fig 5d, the authors suggest that part of the extracellular malectin-like domains is sufficient for FER interaction with LRX4. Could the authors include a brief discussion on this result in light of previous hypotheses (summarized by Li, Wu, and Cheung, 2016) that the malectin domains of FER and related RLKs themselves may interact with carbohydrates in the cell wall and contribute to sensing.

> Thank you once again for your suggestion to relate our work to a broader perspective. In the revised version of the manuscript we include a brief discussion on FER binding to carbohydrates/pectin. Based on our and previous work, we assume that the malectin domain of FER binds to various interactors/molecules, such as RALFs, LRXs or possibly pectin. Recent structural insight into the tandem malectin domain, however, questions its binding to carbohydrates and favours a protein-protein interaction scenario.

Additional minor comments:

3) Line 121: no section F in Fig 3

> Amended

4) Line 132 - 138: Co-expression data with FER is only shown for LRX3. Is expression data similar for LRX4 and LRX5? Are these the only three LRXs with spatial expression of interest (i.e., expressed in root)?

> We updated our analysis, showing that perturbations induce transcriptional changes in several *LRX* genes as well as in *FER*. Notably, the correlation is most overlapping with LRX3, which we consequently also used for a more detailed co-expression analysis.

5) Since the triple mutant *lrx3 lrx4 lrx5* is used for the cell wall sensing experiments, I assume the three genes are likely functionally redundant? In the discussion, the authors may wish to briefly comment on this redundancy and what it might mean for function in FER-dependent cell wall sensing.

> We indeed observe functional redundancy among these three genes. We further elaborated on the functional redundancy between LRX3-5 and dedicated a novel Supplementary figure to this matter.

The manuscript is well written with questions, hypotheses, and data clearly presented.

> Thank you for your constructive suggestions, which we gratefully integrated to further improved our manuscript!

Referee #2:

This manuscript by Dunser et al. reports the measurements of vacuole size/morphology in Arabidopsis root cells to demonstrate some genetic components required for maintaining the normal vacuole development. They have chosen to examine whether genetic mutation or overexpression of a receptor like kinase FERONIA, a family of potential signaling proteins localized in extracellular side of the plasma membrane, leucine-rich-repeat (LRR) extensin, and auxin responsive signaling

protein, SAUR19 as those proteins were previously established to be some of the perhaps many critical components of cell growth by other laboratories. In this study, the authors have developed 3D imaging techniques with the use of organelle-specific fluorescent dyes in order to visualize the vacuole, quantify vacuolar phenotypes, and compare wildtype and those mutants.

They also examined how various compounds affect vacuolar size by treating Arabidopsis roots with several reagents including a PM proton pump stimulator, fusicoccin, a growth inhibitory peptide, RALF, and a pectin methylesterase inhibitor, EGCG. From those experiments, they conclude that FERONIA as well as LRR-extensin that were previously known to be involved in cell expansion process are also involved in determining vacuole size/morphology. Overall they provide interesting observations on the correlation of FER and LRR-extensin mutations with vacuole morphology. I have various comments to this manuscript which the authors may consider, to improve the manuscript before publication to the EMBO J.

> Thank you for help to further strengthen our manuscript.

Whereas it is nice that they can find the correlation of vacuole size and the predicted role of PM signaling protein FER and LRR-extensin, in cell expansion, I (probably a majority of readers) would like to see some models for the connection of the cell surface events and vacuole.

> In the revised version of the manuscript we show that apoplast acidification has a rather fast impact on vacuolar shape (within 30 minutes of FC treatment). Therefore, we assume that some FER signalling components have a rather direct impact on some vacuolar effectors, which we intend to further investigate in the near future. In the manuscript, we also discuss that FER-dependent control of the actin cytoskeleton could possibly impact on vacuolar morphology.

All the datasets they show are correlative responses under growth-altering physiological or genetic conditions. There are some known mutants involved in vacuole and cell growth (such as V-type H-ATPase, Genes Dev. 1999 Dec 15; 13(24): 3259-3270., The Plant Cell, Vol. 26: 3416-3434 or the original paper which described H⁺ pyrophosphatase defective phenotype). They could mention some mutant phenotypes describing defective function in the vacuole and cell growth in the Discussion section if their experimental results are not available.

> The multifunctional vacuole has an essential role in plant development. Its currently unknown if and how the mentioned acidification of the vacuole is related to the size regulation of the vacuole. Moreover, the acidic nature of the vacuole is also important for the pH homeostasis of the cytosol, which could independently of the size impact on cellular growth. We started to look into this topic together with the Schumacher lab, but the topic would exceed the scope of this manuscript. We, however, followed the valuable suggestion and provided a better introduction into the importance of the vacuole for plant growth and development.

Clearly there are going to be MANY movements of various organic and inorganic ions between the extracellular fluid, the cytoplasmic fluid and the vacuolar fluid and thus these three compartments are going to see many fluxes in all directions all contributing to the longer term morphological changes they are reporting for the vacuole. Time courses for things like FC and RALF, that act in seconds and a few minutes, rather than hours, would be a tremendous improvement.

> We thank you for the suggestion and performed time course experiments using FC treatments. We observed an impact of FC on apoplastic pH and vacuolar morphology within 30 minutes. We accordingly conclude that the influence of apoplast acidification on vacuolar size is rather rapid.

Line 145, they mention that salt stress damage the cell wall, but no mention about negative effects of high salts on the plasma membrane nor cytoplasm. Some Salt Overly Sensitive mutants are defective in the plasma membrane transporter or localized at the plasma membrane. They may not assume the negative effect of high salts are only at the cell wall.

> Thank you for this suggestion. We added a related comment to this paragraph.

Line 212, they state FER-kinase function is required for FER phosphorylation events and cite Ref. 24. In fact, Ref 24 did not demonstrate the kinase dependent role of FER. The three papers that show

kinase-dependent role of FER are "Receptor kinase complex transmits RALF peptide signal to inhibit root growth in Arabidopsis, PNAS", "The Receptor-like Kinase FERONIA Is Required for Mechanical Signal Transduction in Arabidopsis Seedlings, Current Biology", and "Comparison of the effects of a kinase-dead mutation of FERONIA on ovule fertilization and root growth of Arabidopsis, FEBS Letters". Those papers should be referred in line 212 instead of Ref.24.

> We appreciate your help to improve our manuscript. We added the references as suggested.

In Supplemental Fig. 7, it would make sense to include the results of LRR4 overexpressing line with Col-0, fer-4, and lrx3/4/5.

> We could not perform the requested experiment, because the Citrus tag cannot be combined with the HPTS analysis as the fluorescent properties are overlapping.

They have done great 3D imaging and surface rendering of the vacuole in various different root cell types shown in Fig.1. To me they should be using this 3D assay for all the rest of image analyses shown in Fig. 2, 3, 4 and 6. In the images shown in Fig. 2, 3, 4 and 6 are captured in 2D, which potentially induce bias when different planes of the tissues are scanned, unless they provide what the z depth in the roots these planes are examined.

> Notably, both our 3D as well as our 2D experiments were highly reproducible. Single sections for the 2D analysis were taken above the nucleus, indeed ensuring that we inspect a similar plane. We performed both 2D and 3D imaging for the used mutants (pER8:SAUR19, fer, lrx3lrx4lrx5) and treatments (FC, acidic medium, stiff medium, EGCG) in Figure 1, 2, 3, 4 and previous SFigure 2, reflecting the strong correlation between our 2D and 3D data. Afterwards, we have used 2D imaging for example to report resistant vacuolar appearance to EGCG in fer or lrx triple, which we believe is a highly suitable approach. Accordingly, the majority of our analysis is actually represented by 2D and 3D analysis. As other readers may also miss this point, we decided to shift the data of SFigure 2 into the main figure category, enhancing the visibility and representation of our 3D data,.

In the method section of 3D reconstruction of vacuoles, they provide very little information as to the imaging method. What is the pinhole value (either airy unit or absolute value, i.e., nm or micron)? How many optical slices were captured for z stack? They mention step size but no mentioning the number of slices. This number would be useful so that one can get some ideas on the thickness/volume of their images. Please provide either laser power transmission % value of excitation and/or gain setting of confocal imaging. Those would tell us what levels of fluorescent signals were obtained from their specimen.

> We improved the description as requested.

Fig. 2 figure caption heading indicates "with cell wall acidification". But there are no data of pH measurements in this figure. Please correct this.

> We improved the figure caption.

Supplemental Fig 4. There is gene expression profile of THE1. No description of this figure in the manuscript text. Either remove the chart or please describe it.

> We improved our discussion on THE1 (see also comments to reviewer #1)

They treated Arabidopsis roots with fusicoccin, acidic pH, pectin methylesterase inhibitor EGCG or induced SAUR19 by inducible expression using estradiol, to examine the effects of those physiological conditions. The timings of incubation with the conditions vary from 2.5 hours to 22 hours. What we see are the end point result. Ideally it should have done with time-course or real-time measurement (see earlier comment). If possible provide some RAPID time course results, or I would suggest to add some cautionary notes on those endpoint measurements in relation to whether the observations are direct or indirect responses.

> We thank the reviewer for this suggestion and performed time course experiments of vacuolar shapes, which we moreover combined with HPTS imaging to monitor apoplastic acidification. We show that FC treatments modulated both the apoplastic pH and the vacuolar morphology within 30 minutes, suggesting a rather direct impact.

They propose cell wall sensing and vacuolar morphology/size in the context of cell expansion. In all the images in Fig.2, 3, 4, and 6, they conclude vacuole morphology is affected by genetic mutations or chemical treatment. But these cells do not appear to be affected in terms of cell length or width. Are they proposing that vacuole morphology changes before cells expand? Though incubation time is rather long (hours); does one expect to see some changes in cell length or width?

> We indeed are looking at epidermal cells at the end of the meristem to investigate vacuolar dynamics before the onset of major cellular elongation. However, most of the used conditions are indeed expected to quantifiably impact on cellular elongation rates (see also Barbez et al., 2017).

The pulldown experiments are fairly meaningless since these have been done already in several publications and more important, since the proteins contain hydrophobic transmembrane domains and require detergents to be released from the membrane into various types of macromolecular complexes without proper controls (both positive and negative) it is impossible to evaluate the importance of these results.

> There might be a confusion. Besides using the full length of FER (including the transmembrane domain), we also illustrate that part of the Malectin domain of FER (without transmembrane domain) and the LRR domain of LRX interact *in planta*. Moreover, these truncated domains also interacted when heterologously expressed in yeast (two hybrid approach without any extraction). In combination with our genetic data, we conclude that FER and LRX proteins directly interact via their malectin and LRR domains.

Referee #3:

1. It is known from classical studies on plant cytology that plant cells expand through increasing the size of the vacuole as the cell surface - membrane and wall - increases in area. Here the authors describe the change in vacuolar size in root epidermal cells (atrachoblasts) during expansion. They show that the vacuole occupies approximately 98% of the volume of the cell by the end of elongation and that the cytoplasm does not increase appreciably in volume.

2. It is known that cell wall acidification in growing cells can increase cell elongation. The authors describe how three different treatments known to cause changes in cell elongation also resulted in a change in vacuole morphology and the relative volume occupied by the vacuole (vacuolar occupancy). As cells elongate we would expect vacuolar occupancy to increase because all growth is due to increase in vacuolar volume while cytoplasmic volume is more or less constant. Similarly, treatments that reduce cell size would result in a lower vacuolar occupancy volume. This is exactly what the authors show, which is good. This is a nice introductory/confirmatory piece of data.

3. FER has been suggested to be involved in cell wall mechanical sensing. Vacuolar volume of *fer* is different from wild type. The authors suggest that this indicates that FER regulates vacuolar volume. How do they know that the effect of the *fer* mutation does not cause a defect in cell expansion/cell size which in turn impacts vacuolar volume? To make their point they should include measures of cell size in wild type and *fer* mutants. If their assertion is correct, then wild type and *fer* atrichoblast cells must be the same size. These data are not in the paper at the moment.

> The median vacuolar occupancy of *fer* and *lrx3 lrx4 lrx5* mutants is above 60% (see Figure 3B and 4B). Such a dilation of the vacuole is associated with a substantial increase in cell size in wild type conditions (see Figure 1A-C), which is certainly not apparent in late meristematic *fer* and *lrx3 lrx4 lrx5* mutant cells. We measured the cell length of *fer* and *lrx3 lrx4 lrx5* mutants as suggested and detected a tendency (not always statistically significant) for longer cells in *fer*, but not *lrx3 lrx4 lrx5*, mutants. We conclude that the vacuolar enlargement in *fer* and *lrx3 lrx4 lrx5* cannot be solely explained by larger cells in this background.

4. LRX proteins are required for cell growth and thought to play structural roles in the cell walls. The *lrx* triple mutant develops defective vacuoles compared to wild type. It would be good if the authors should include the measure of *lrx* trip mutant atrichoblasts to ensure that the effect that is observed is not a result of the differences in cell size between the different genetic backgrounds.

> Please, see our answer above.

5. The authors demonstrate that LRX and FER proteins interact.

Major points:

Cell sizes need to be included as outlined under points 3 and 4 above to ensure that the changes on vacuolar morphology are not an indirect effect of cell size. I assume that they authors have these data so this should not be a big deal.

> We performed the experiment as suggested and conclude that the vacuolar enlargement in *fer* and *lrx3 lrx4 lrx5* cannot be solely explained by larger cells in this background.

Minor points:

The writing style is poor in places. It would be good if the authors could get editorial comments from someone outside the field. Listed here are some suggested changes, but there are many more.

> Thanks for the suggestion. We revised our manuscript according to the suggestions.

Line 77; define what you mean by vascular occupancy in the Results section.

> We measured how much space the vacuole fills within its cell. This we now clearly define as “vacuolar occupancy of the cell” in the revised version of the manuscript.

Line 104-105; This suggests that the effect is causative. it could be a secondary effect. the authors should change their wording

> We changed our wording as suggested.

Line 112: The authors should demonstrate or mention that EGCG-treatment resulted in less elongation in their hands.

> We followed your suggestion and added root growth inhibition assays to the revised version of the manuscript.

Line 120; *fer* mutants showed enlarged vacuoles compared to what. The authors should state this in the sentence.

> Amended.

Line 136; What does interesting mean?

> Amended.

Line 139 than in wild type.

> Amended.

Line 170; Could it be that FER has a structural role like LRX? These data are consistent with that hypothesis. They provide no support for a signaling role of FER.

> This is an interesting hypothesis. However, our data also proposes that the kinase domain of FER is required to restrict vacuolar expansion. Hence, we assume that FER RLK-dependent signal transduction is involved.

Line 183: delete "remarkably". It makes it look as though the authors are trying to oversell their data/results.

> Amended as suggested.

2nd Editorial Decision

20th December 2018

As you will see they both find that all criticisms have been sufficiently addressed and recommend the manuscript for publication. However, before we can officially accept this study there are a few editorial issues concerning text and figures that I need you to address.

REFEREE REPORTS

Referee #1:

In my opinion, the authors have satisfactorily addressed the reviewer comments. Amendments to the introduction and discussion strengthen the integration of this work with the broader plant development and RLK fields. Functional redundancy of the LRX genes is more specifically addressed and greater clarity provided for readers.

Referee #2:

They revised the MS accordingly and added appropriate pictures. I recommend acceptance.

Corresponding Author Name: Jürgen Kleine-Vehn

Manuscript Number: EMBOJ-2018-100353R